# Tumor budding score predicts lymph node status in oral tongue squamous cell carcinoma and should be included in the pathology report

Inger-Heidi Bjerkli[1,2], Helene Laurvik[3], Elisabeth Sivy Nginamau[4,5], Tine M. Søland[3,6], Daniela Costea[4,5], Håkon Hov[7,8], Lars Uhlin-Hansen[2,9], Elin Hadler-Olsen[2,10], Sonja E. Steigen[2,9]*

1 Department of Otorhinolaryngology, University Hospital of North Norway, Tromsø, Norway, 2 Department of Medical Biology, UiT The Arctic University of Norway, Tromsø, Norway, 3 Department of Pathology, Rikshospitalet, Oslo University Hospital, Oslo, Norway, 4 Department of Pathology, Haukeland University Hospital, Bergen, Norway, 5 Department of Clinical Medicine, Center for Cancer Biomarkers CCBIO and The Gade Laboratory of Pathology, University of Bergen, Bergen, Norway, 6 Faculty of Dentistry, Institute of Oral Biology, University of Oslo, Oslo, Norway, 7 Department of Pathology, St. Olavs Hospital, Trondheim, Norway, 8 Department of Clinical and Molecular Medicine, Faculty of Medicine and Health Sciences, Norwegian University of Science and Technology, Trondheim, Norway, 9 Department of Pathology, University Hospital of North Norway, Tromsø, Norway, 10 The Public Dental Health Service Competence Center of Northern Norway, Tromsø, Norway

* Sonja.eriksson.steigen@uit.no

**Data Availability Statement:** All relevant data are within the manuscript and its Supporting Information files.

## Abstract

### Background

The majority of oral cavity cancers arise in the oral tongue. The aim of this study was to evaluate the prognostic value of tumor budding in oral tongue squamous cell carcinoma, both as a separate variable and in combination with depth of invasion. We also assessed the prognostic impact of the 8th edition of the American Joint Committee on Cancer's TNM classification (TNM8), where depth of invasion (DOI) supplements diameter in the tumor size (T) categorization.

### Methods

Patients diagnosed with primary oral tongue squamous cell carcinoma were evaluated retrospectively. Spearman bivariate correlation analyses with bootstrapping were used to identify correlation between variables. Prognostic value of clinical and histopathological variables was assessed by Log rank and Cox regression analyses with bootstrapping using 5-year disease specific survival as outcome. The significance level for the hypothesis test was 0.05.

### Results

One-hundred and fifty patients had available material for microscopic evaluation on Hematoxylin and Eosin-stained slides and were included in the analyses. Reclassification of

**Funding:** Inger-Heidi Bjerkli has received financial support for this work from Helse Nord (Nonprofit organization). The funder had no role in planning or execution of the study, nor in interpretation of results or writing of the manuscript.

**Competing interests:** The authors have declared that no competing interests exist.

tumors according to TNM8 caused a shift towards a higher T status compared to the previous classification. The tumor budding score was associated with lymph node metastases where 23% of the patients with low-budding tumors had lymph node metastases, compared with 43% of those with high-budding tumors. T-status, lymph node status, tumor budding, depth of invasion, and the combined tumor budding/depth of invasion score were all significantly associated with survival in univariate analyses. In multivariate analyses only N-status was an independent prognosticator of survival.

## Conclusion

Reclassification according to TNM8 shifted many tumors to a higher T-status, and also increased the prognostic value of the T-status. This supports the implementation of depth of invasion to the T-categorization in TNM8. Tumor budding correlated with lymph node metastases and survival. Therefore, information on tumor budding can aid clinicians in treatment planning and should be included in pathology reports of oral tongue squamous cell carcinomas.

## Introduction

Oral cavity cancer is the most common subtype of head and neck cancer [1], and squamous cell carcinomas (SCC) account for about 90% of these [2, 3]. The majority of oral cavity cancers arise in the mobile, anterior two-thirds of the tongue called the oral tongue [2]. Recent studies report rising incidence of oral tongue (OT)SCC, especially in younger patients [4–6].

Patients with OTSCC have a high morbidity and poor prognosis even when tumors are small [7]. For low-stage tumors (T1-T2N0M0), as much as 20% of patients may develop neck metastasis and recurrence within 2 years [8, 9]. In Europe the five-year relative survival for oral cancer is around 50% [10]. Surgical removal of the tumor is the preferred choice of treatment for OTSCC in most institutions when the tumor is regarded resectable. In addition, neck dissection is performed when positive lymph nodes are detected clinically. For patients with clinically negative lymph node status, there is no established biomarker or method to predict whether they will benefit from neck dissection. Postoperative radiation therapy is recommended when the histopathological examination reveals short tumor margins and/or a positive lymph node status. Chemotherapy is mostly used for palliative treatment [11, 12].

Squamous cell carcinomas are classified according to the tumor's greatest diameter and depth of invasion (T), regional lymph node metastases (N), and organ metastasis (M), according to the TNM system [2, 3]. The TNM classification is the most established predictor of patient survival in OTSCC, but it gives limited information about the aggressiveness of the tumor. The latest edition of the TNM classification (TNM8) has been in use since 2017. In this edition the depth of tumor invasion (DOI) supplements tumor greatest diameter in the T categorization, in an attempt to increase its prognostic value [2, 13]. Prior to the TNM8 classification, only greatest diameter was included in the evaluation of T, and DOI was proposed as a separate prognostic variable [14]. Some report that the new T criteria in TNM8 lead to a shift to higher T-status compared with previous TNM editions, those who remain in a lower T-status have better survival outcomes when DOI is included [15].

Other histopathological characteristics than DOI have also been evaluated for prognostic value, such as tumor budding (TB). TB is defined as invading clusters of four or fewer tumor

cells at the invasive front, and has been associated in many studies with lymph node metastasis, relapse, and accordingly poor prognosis [16–22]. In many studies, TB has been proposed to be a useful and significant prognostic marker that can be evaluated on hematoxylin and eosin (H&E)-stained sections, at low cost and with fair reproducibility [18, 23–25]. After validation of the International Tumor Budding Consensus Conference (ITBCC 2016) TB is suggested to be included in the routine histopathological report for oral cancers [21, 22, 26]. An additional prognostic factor, a combined score of TB and DOI, was suggested for early stages of OTSCC, before the DOI was implemented into the TNM8 [14, 27].

The aim of this study was to evaluate the prognostic impact of the T classification after implementation of DOI as described in TNM8, in a large cohort of OTSCC. We also assessed the prognostic value of TB-scores and DOI as separate variables, as well as in combination as a TB/DOI score.

## Methods

### Identification of patients

In Norway, management of oral cavity cancer is centralized to the university hospitals of Rikshospitalet (Oslo), Haukeland (Bergen), St. Olavs (Trondheim) and North Norway (Tromsø). The Norwegian oral cancer (NOROC) study is a retrospective study that includes the majority of patients diagnosed with oral cavity SCC in Norway from January 1st 2005 through December 31st 2009 [28]. Patients were identified by searching for the relevant ICD-10 codes in the electronic health record, and by searching the hospital's pathology archives for cancers with topographic SNOMED coding T51 and T53, which were subsequently matched with the relevant ICD-10 codes recorded in the electronic health records. In this sub-study, the relevant ICD-10 code was C02, which refers to cancers in the mobile tongue. Patients with cancers other than SCC were excluded, as well as patients with second primaries or previous cancer treatment, and patients from whom formalin fixed, paraffin embedded tumor tissue was lacking.

### Extracting clinical data

Relevant patient data including age, gender, ICD-10 diagnosis, TNM classification, treatment, and follow-up were registered as previously described [28]. Cause of death was acquired from Norwegian Cause of Death Registry. Patients were divided into ten-year interval groups (51–60 years, 61–70 years, and 71–80 years) based on age at time of diagnosis. Patients younger than 50 years and older than 80 years were pooled in younger ($\leq$50 years) and older ($\geq$81 years) age groups. Survival was measured from the date of diagnosis until death or last day of follow-up, which was June 1st 2015 ensuring a minimum of five years of follow-up for surviving patients.

### Histological samples and categorical grouping

TB and DOI were scored on H&E-stained sections by calibrated and experienced pathologists (HL, ESN, TMS, DC, HH, LUH and SES). No special stains were provided. The scoring was done independently or by pairs of pathologists who were blinded for the patients'clinical outcomes. Biopsies or resections that were too fragmented, too shallow, too superficial, or with technical artefacts that rendered the histological evaluation impossible, were excluded; thus the number of cases with information of TB and DOI varies.

The TB was assessed after scanning 10 separate fields along the invasive front before counting number of buds in the single field (20x objective) with the highest density (hotspot) [26].

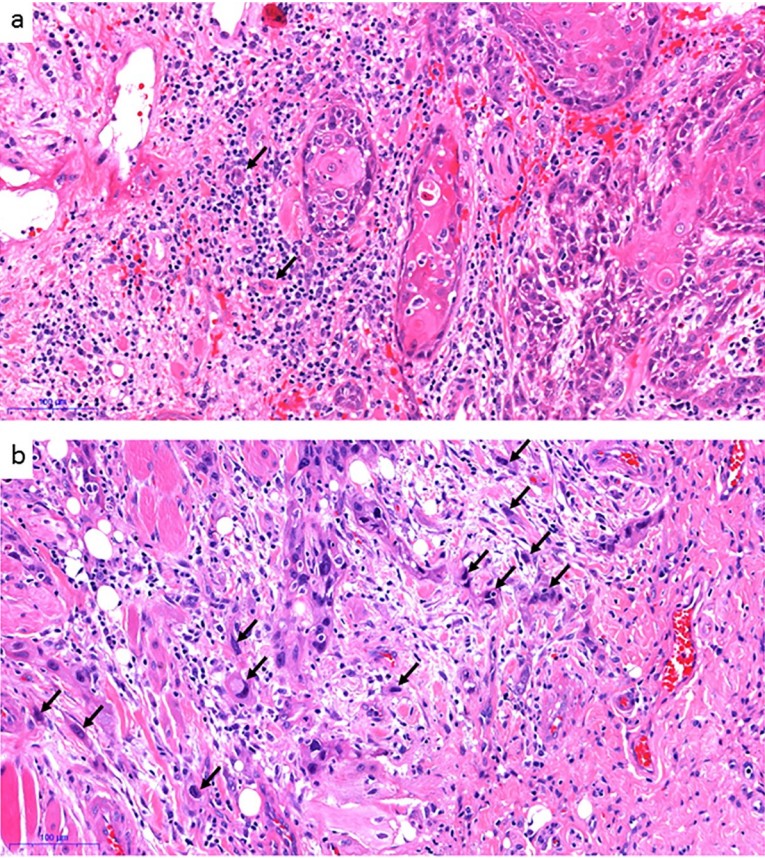

**Fig 1. Tumor budding.** A tumor with low number of buds (marked with black arrows) shown in a, and a tumor with high number of buds in b.

TB was categorized into a two-tier (2-tier) system where 0 through 4 buds were denoted low-Bd and $\geq$ 5 buds high-Bd, according to the work of Wang et al. and Xie et al. [17, 29] (Fig 1) Also, the three-tier (3-tier) system recommended by the International Tumor Budding Consensus Conference was applied, where 0 through 4 buds is denoted Bd1, 5 through 9 buds as Bd2, and 10 and more buds as Bd3 [26].

DOI was measured as the perpendicular distance from the basement membrane region to the deepest point of the infiltrative front of the tumor in millimeters, and divided into a DOI 2-tier system with cutoff $\geq$4mm according to Almangush et al. [23] as well as into a 3-tier system according to TNM8: $\leq$5 mm, 5.1–10.0 mm, and >10 mm [2].

The combined score of low or high number of buds and DOI was assigned according to Almangush et al. [14, 27]. In short, tumors with < 5 buds and DOI < 4 mm were given TB/DOI-score 0. Tumors with either < 5 buds and DOI$\geq$4 mm, or $\geq$5 buds and DOI< 4 mm were given TB/DOI-score 1, whereas tumors with $\geq$5 buds and DOI $\geq$4 mm were given TB/DOI-score 2.

## Ethics

The study was approved by the Institutional Review Board of the Northern Norwegian Regional Committee for Medical Research Ethics (REK Nord) (Protocol number REK Nord; 2013/1786 and 2015/1381), applicable to all four hospitals. A patient information-consent

**Table 1. Shift in T-classification; T-status according to old classification in the first column, the T-status according to TNM8 in the second.**

| T-status according to old classification | T-status according to TNM8 classification | Number of cases |
|---|---|---|
| 1 | 1 | 43 |
| 1 | 2 | 31 |
| 1 | 3 | 17 |
| 2 | 1 | 1 |
| 2 | 2 | 17 |
| 2 | 3 | 10 |
| 3 | 2 | 1 |
| 3 | 3 | 6 |
| 4 | 3 | 1 |

Coincident numbers indicates no changes in T-status while discrepancy indicates a shift from a lower to a higher T-status, or from a higher to lower (only two cases). The third column indicates the number of cases for each of the alternatives.

letter was sent to the patients still alive at the start of the retrospective study, giving them the opportunity to opt-out of the study.

## Statistical analyses

Descriptive statistics with frequencies were used to describe the cohort. Bivariate correlation between variables was assessed by Spearman correlation analyses including bootstrapping (Tables 2 and 3). Univariate survival analyses were conducted using the Log-rank test. Cox regression (proportional hazard regression) with bootstrapping was applied for calculating survival analyses, hazard ration and 95% confidence intervals (Tables 4 and 5). Collinearity was evaluated with linear regression. Multivariate survival analyses were conducted using forward-stepwise Cox regression. The significance level for the hypothesis test was 0.05 [30]. Statistical analyses were performed with IBM Statistical Package for the Social Sciences (SPSS) statistics, version 26.

## Results

### Clinical characteristics

Altogether 239 patients with primary, treatment-naïve OTSCC were identified, and 200 (84%) of these were included in the study as they received treatment with curative intent. For the

**Table 2. Spearman bivariate correlation between TB and the variables T, N and DOI in the whole cohort.**

|  | T old | T 8 ed | DOI 2-tier | DOI 3-tier | N |
|---|---|---|---|---|---|
|  | n = 126 | n = 132 | n = 130 | n = 130 | n = 139 |
| **TB 2-tier** | r = 0.009 | r = 0.050 | r = 0.074 | r = 0.075 | r = 0.210 |
|  | CI: -0.152–0.188 | CI: -0.108–0.212 | CI: -0.094–0.221 | CI: -0.103–0.238 | CI: 0.033–0.392 |
|  | p = 0.919 | p = 0.566 | p = 0.403 | p = 0.397 | p = 0.013* |
| **TB 3-tier** | r = 0.003 | r = 0.043 | r = 0.070 | r = 0.066 | r = 0.185 |
|  | CI: -0.172–0.179 | CI: -0.124–0.199 | CI: -0.107–0.211 | CI: -0.107–0.220 | CI: -0.052–0.358 |
|  | p = 0.975 | p = 0.624 | p = 0.429 | p = 0.459 | p = 0.030* |

n = number of cases

r = Spearman's Rho

CI = Confidence interval

p = significance (two-tailed)

**Table 3. Spearman bivariate correlation between DOI 2-tier, DOI 3-tier and TB/DOI and the variables T and N in the whole cohort.**

|  | T old | N |
|---|---|---|
|  | n = 124 | n = 131 |
| **DOI** | r = 0.218 | r = 0.181 |
| **2-tier** | CI: 0.084–0.335 | CI: 0.041–0.302 |
|  | p = 0.015* | 0.038* |
|  | n = 124 | n = 131 |
| **DOI** | r = 0.425 | r = 0.241 |
| **3-tier** | CI: 0.282–0.549 | CI: 0.100–0.384 |
|  | p<0.001* | p = 0.024* |
|  | n = 122 | n = 129 |
| **TB/DOI score** | r = 0.147 | r = 0.262 |
|  | CI: 0.002–0.297 | CI: 0.104–0.419 |
|  | p = 0.105 | p = 0.003* |

n = number of cases.

r = Spearman's Rho

CI = Confidence interval

p = significance (two-tailed)

remaining 39 patients, palliative treatment was recorded for 16 (6.7%), and information was missing for 23 (9.6%); these patients were excluded. During the five-year follow-up time, 37 (18.5%) patients developed local recurrence and 23 (12%) patients developed a second primary head and neck cancer. H&E-stained tumor sections were available for 150 (75%) of the cases, of which 127 (84.7%) were resections, 18 (12%) were biopsies, and 5 (3.3%) lacked information.

When reclassifying T-status from the older TNM (T old) to the TNM8 edition (T 8 ed), 31 tumors shifted from T1 to T2, 17 shifted from T1 to T3, and 10 shifted from T2 to T3 (Table 1). Only two tumors shifted to a lower T status.

## Correlation analyses

Univariate correlation analyses were performed on the whole cohort to assess if tumor budding (2 and 3 tier) and depth of invasion were significantly correlated with each other and with, T-status (old and new) or lymph node status (N). TB score (both 2-tier and 3-tier) correlated with lymph node status (Table 2).

42.8% of tumors with a high TB-score had metastasized to lymph nodes, compared with 22.5% of the tumors with low TB-score. For patients with low-stage disease (T1-T2N0M0), none of the analyzed variables were correlated with TB-score.

DOI (2-tier and 3-tier) and the combined TB/DOI score were significantly correlated with T old (tumor diameter) and lymph node status (N) (Table 3), whereas TB/DOI-score was only significantly correlated with lymph node status (N).

For low-stage disease, only DOI 3 tier was associated with tumor (T old) diameter (p<0.001, CI: 0.234–0.605, r = 0.431).

## Univariate survival analyses

In contrast to T-status according to the old TNM edition, the T-status in line with the TNM8 edition was significantly associated with DSS for the whole cohort, as shown in Table 4. TB

**Table 4. Characteristics of the patients (n = 150) with OTSCC treated in curative intent, including number of cases, percent of patients with disease-specific survival (DSS) with p-value (p), 95% confidence interval (CI), and hazard ratio (HR).**

| Characteristic | | No. of cases (%) | 5-year DSS % | HR (CI) p |
|---|---|---|---|---|
| **Gender** | Male | 92 (61.3) | 70.7 | 1.130 (0.622–2.051), p = 0.702 |
| | Female | 58 (38.7) | 69.0 | |
| **Age (year) group** | ≤ 50 | 29 (19.3) | 79.3 | 1.382 (1.072–1.782) p = 0.015 |
| | 51–60 | 29 (19.3) | 65.5 | |
| | 61–70 | 44 (29.3) | 70.5 | |
| | 71–80 | 31 (20.7) | 67.7 | |
| | ≥81 | 17 (11.3) | 64.7 | |
| **T old classification** | T1 | 91 (60.7) | 70.3 | 1.063 (0.666–1.696) p = 0.806 |
| | T2 | 28 (18.7) | 75.0 | |
| | T3 | 8 (5.3) | 75.0. | |
| | T4 | 2 (1.3) | 50.0 | |
| | T unknown | 21 (14.0) | | |
| **T 8 ed classification** | pT1 | 48 (32.0) | 87.5 | 1.734 (1.132–2.657) p = 0.006 |
| | pT2 | 53 (35.3) | 64.2 | |
| | pT3 | 34 (22.7) | 64.7 | |
| | pT unknown | 15 (10.0) | | |
| **N** | N0 | 108 (72.0) | 82.4 | 4.639 (2.541–8.471) p = 0.001 |
| | N+ | 40 (26.7) | 37.5 | |
| | Unknown | 2 (1.3) | | |
| **TB 2-tier** | Low (Bd1) | 112 (74.7) | 76.8 | 2.269 (1.182–4.356) p = 0.016 |
| | High (Bd2+Bd3) | 29 (19.3) | 51.7 | |
| | Unknown | 9 (6.0) | | |
| **TB 3-tier** | Bd1 <5 | 112 (74.7) | 76.8 | 1.847 (1.242–2.748) p = 0.002 |
| | Bd2 ≥5 and<10 | 18 (12.0) | 66.7 | |
| | Bd3 ≥ 10 | 11 (7.3) | 27.3 | |
| | Unknown | 9 (6.0) | | |
| **DOI 2-tier** | < 4 mm | 29 (19.3) | 86.2 | 2.172 (0.769–6.133) p = 0.090 |
| | ≥4 mm | 103 (68.7) | 68.0 | |
| | Unknown | 18 (12.0) | | |
| **DOI 3-tier** | < 5 mm | 47 (31.3) | 87.2 | 1.634 (1.072–2.491) p = 0.015 |
| | ≥5 mm and<10 mm | 52 (34.7) | 61.5 | |
| | ≥10 mm | 33 (22.0) | 66.7 | |
| | Unknown | 18 (12.0) | | |
| **Combined TB/DOI*-score** | TB/DOI-0 | 24 (16.0) | 91.7 | 1.673 (0.969–2.891) p = 0.013 |
| | TB/DOI-1 | 91 (60.7) | 67.0 | |
| | TB/DOI-2 | 20 (13.3) | 60.0 | |
| | Unknown | 15 (10.0) | | |

* Almangush et al 2014.

was significantly associated with 5-year DSS using both the 2-tier and 3-tier system (Fig 2), whereas DOI was only significantly associated with 5-year DSS when assessed by the 3-tier system.

TB was not a significant prognosticator for DSS when patients with low-stage and high stage disease were analyzed separately. The majority of patients in both the low-stage and high-stage disease group had low-budding tumors (87% and 73% respectively, had <5 buds).

**Table 5. Low-stage and high-stage disease evaluated separately.** Percentage of 5-year DSS and p-value for each group.

| | | Low-stage | | | High-stage | | |
|---|---|---|---|---|---|---|---|
| | | No. of cases (%) | DSS % | HR (CI) p | No. of cases (%) | DSS % | HR (CI) p |
| **T old classification** | T1 | 58 (80.6) | 79.2 | 0.411 (0.053–3.215) p = 0.260 | 32 (57.1) | 45.2 | 0.887 (0.554–1.4174) p = 0.591 |
| | T2 | 14 (19.4) | 90.9 | | 14 (25.0) | 45.5 | |
| | T3 | ÷ | ÷ | | 8 (14.3) | 66.7 | |
| | T4 | ÷ | ÷ | | 2 (3.6)) | 50.0 | |
| **T 8 ed classification** | T1 | 42 (54.5) | 91.2 | 3.344 (0.886–12.616) p = 0.035 | 5 (8,8) | 25.0 | 0.732 (0.425–1.266) p = 0.210 |
| | T2 | 35 (45.5) | 73.3 | | 18 (31.6) | 35.3 | |
| | T3 | ÷ | ÷ | | 34 (59.6) | 58.6 | |
| **N** | **N0** | 77 (100) | 87.5 | ÷ | 23 (36.5) | 70.0 | 2.862 (1.167–7.020) p = 0.021 |
| | N+ | ÷ | ÷ | | 40 (63.5) | 30.6 | |
| **TB 2-tier** | Low (Bd1) | 65 (86.7) | 86.8 | 2.872 (0.742–11.121) p = 0.089 | 43 (72.9) | 48.6 | 1.305 (0.558–2.894) p = 0.496 |
| | High (Bd2+Bd3) | 10 (13.3) | 66.7 | | 16 (27.1) | 40.0 | |
| **TB 3-tier** | Bd1 <5 | 65 (86.7) | 86.8 | 1.875 (0.784–4.489) p = 0.091 | 43 (72.9) | 48.6 | 1.427 (0.890–2.290) p = 0.094 |
| | Bd2 ≥5 and<10 | 7 (9.3) | 66.7 | | 8 (13.6) | 71.4 | |
| | Bd3 ≥ 10 | 3 (4.0) | 66.7 | | 8 (13.6) | 12.5 | |
| **DOI 2-tier** | < 4 mm | 25 (33.8) | 94.4 | 4.415 (0.565–34.505) p = 0.098 | 3 (5.3) | 0.0 | 0.461 (0.138–1.545) p = 0.016 |
| | ≥4 mm | 49 (66.2) | 76.7 | | 54 (94.7) | 51.1 | |
| **DOI 3-tier** | < 5 mm | 41 (53.2) | 90.9 | 3.513 (0.931–13.256) p = 0.043 | 5 (8.8) | 25.0 | 0.686 (0.400–1.175) p = 0.119 |
| | ≥5 mm and<10 mm | 33 (44.6) | 71.4 | | 19 (33.3) | 33.3 | |
| | ≥10 mm | 0 | ÷ | | 33 (57.9) | 60.7 | |
| **Combined TB/DOI*-score** | TB/DOI-0 | 21 (28.8) | 100 | 2.797 (0.997–7.843) p = 0.005 | 2 (3.4) | 0.0 | 0.843 (0.388–1.833) p = 0.609 |
| | TB/DOI-1 | 45 (61.6) | 77.5 | | 44 (74.6) | 50.0 | |
| | TB/DOI-2 | 7 (9.6) | 66.7 | | 13 (22.0) | 50.0 | |

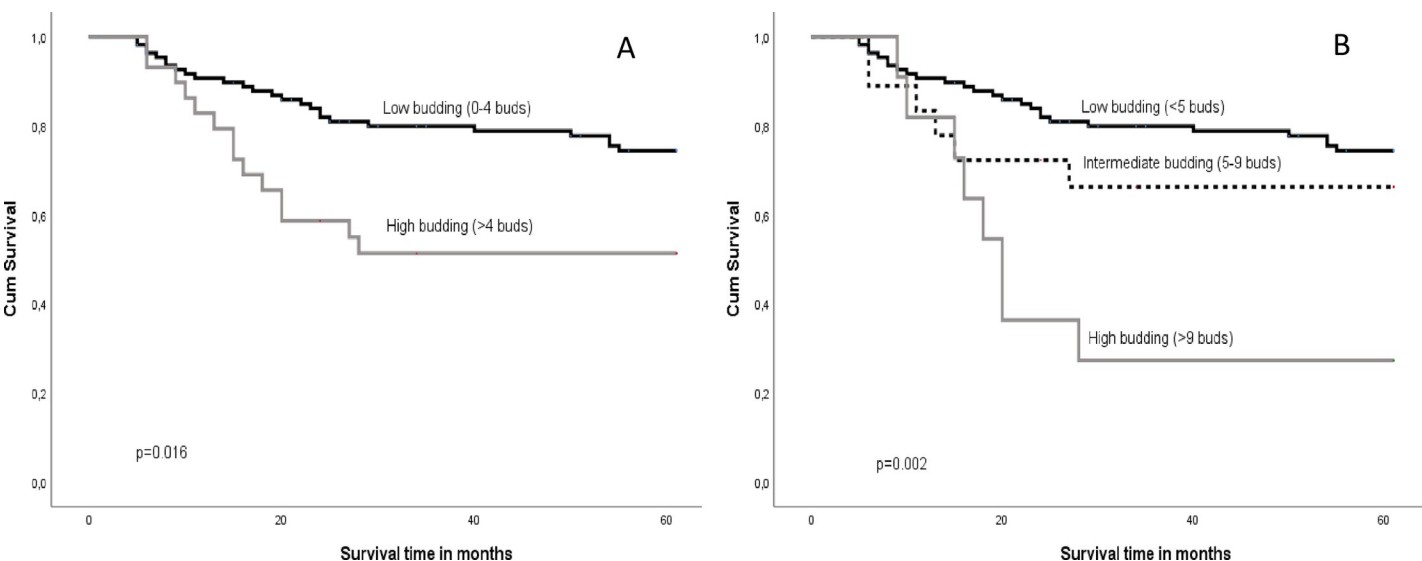

**Fig 2.** Kaplan Meier survival curves based on TB applying the 2-tier cutoff for TB (A) and the 3-tier cutoff (B).

DOI was a significant predictor of 5-year DSS in the low-stage disease group when assessed by the 3-tier scale, and in the high-stage disease group using the 2-tier scale. Furthermore, the combination of TB and DOI was significantly associated with 5-year DSS in the low-stage disease group (Table 5).

## Multivariate survival analyses

All variables that were significantly associated with survival in the univariate analyses, were potentially eligible for multivariate cox regression analyses. However, T 8 ed and DOI were highly collinear, and also TB 2-tier and TB 3-tier. Two different equitation's were performed; one with TB-2-tier and one with TB 3-tier. The other histopathological variables implemented in the multivariate survival analyses were T 8 ed and N. Only N was an independent prognosticators of 5-year DSS (p<0.001). For the low-stage disease group only the two variables T 8 ed and the combined TB/DOI-score were significant in univariate tests, but as they were collinear no multivariate analyses were performed.

## Discussion

In this study the prognostic value of the T-status according to the TNM8 edition, where DOI is included, has been evaluated. Also, the prognostic value of TB, DOI, and combined TB/DOI scores has been assessed in a large cohort of primary treatment-naïve OTSCC. There are recent studies that support a prognostic value of these variables in OSCC and validation in a large, homogenous cohort can facilitate clinical implementation of the markers.

The TNM classification system of a cancer does not always provide adequate information for treatment stratification and prognostication. A consequence of this is the risk of overtreatment or undertreatment. Therefore, it is important to find a reliable and reproducible method to distinguish between aggressive OTSCC needing more extensive treatment, such as neck dissection surgery and postoperative radiotherapy, and less aggressive OTSCC, where the patients can be spared from the burden of the latter treatment modalities. The use of simple prognostic markers or parameters that can be easily assessed on H&E-stained histological sections from tumor biopsies or resection specimens is an ideal approach, such as the assessment of DOI and TB. They were analyzed without the need of expensive equipment, reagents, or extra laboratory procedures, in accordance with other studies [14, 19, 21, 22, 24, 27].

The DOI 3-tier showed a significant positive correlation with T old and N-status, supporting the introduction of tumor DOI to the T-status. When we re-classified the T-status according to TNM8 where DOI is included, there was a shift towards a higher number of T2 and T3 tumors, which is in line with a previous study [15]. The newest T-classification was a significant prognosticator for 5-year DSS in univariate analyses, in contrast to T according to the old classification that only included diameter of the tumor.

Several studies have investigated TB in low-stage OTSCC (T1-T2N0M0), where TB has been correlated with lymph node metastasis and poor prognosis [14, 29]. In some studies, a low TB count correlated well with longer survival, and TB has been suggested to be a valuable prognostic marker for OTSCC that should be implemented in treatment decision-making [21]. TB was not a significant prognostic marker in the present study. There may be several reasons for the discrepancy between our study and the studies by Almangush and coworker [14] and Xie and coworker [29]. The number of patients with low-stage disease was lower in the present study compared to the studies by Almangush and Xie. Similar to these previous studies we found that high-budding tumors were associated with shorter survival than low-budding tumors, but the differences were not statistically significant. Although a reasonable number of patients were included in the present study, splitting the cohort into subgroups

may have caused sub-group analyses to be under-powered. This is a limitation to our study. The percentage of high-budding tumors was lower in the present study compared to the studies by Almangush and coworkers and Xie and coworkers, which may also have affected the results. In a study by Manjula and coworkers, TB was also not an independent prognostic marker in multivariate analyses [24]. However, this study used 10 buds as cut-off between high- and low-budding, thus the results cannot be directly compared to the present study where we used other cut-off points.

Degree of budding was associated with lymph node metastases as stated by others [31]. This could be of clinical importance for patients where the clinicians have restrained from neck dissection. If the pathology report states a high degree of budding this might indicate a higher chance of lymph node metastases, implying that a tighter follow up is warranted. This could include new radiological imaging for evaluation of lymph nodes at short intervals [32], or neck dissection when in doubt. Our results give no clear indication of whether the TB 2-tier or 3-tier scoring is best. None of the TB scoring systems were independent prognosticators of survival in our study. The correlation to lymph node metastases was stronger for the TB 3-tier system than for the 2-tier system. In univariate survival analyses the hazard ratio was larger using the TB 2-tier system than for the 3-tier system, but with a larger confidence interval and a higher p-value. The Kaplan Meier curves also demonstrates that the high budding group according to the 3-tire system has particularly poor survival, suggesting that patient in this group may require more extensive treatment and a closer follow-up than patients with low-budding tumors. Taken together these results suggest that the TB 3-tier system has better prognostic power than the 2-tier system, but further studies are needed to draw definite conclusions.

## Conclusion

Reclassification according to TNM8 shifted many tumors to a higher T-status, and also increased the prognostic value of the T-status. This supports the implementation of depth of invasion to the T-categorization in TNM8. Tumor budding was associated with lymph node metastases and survival. Therefore, information of tumor budding can aid clinicians in treatment planning and should be included in pathology reports of oral tongue squamous cell carcinomas.

## Supporting information

**S1 SPSS dataset.**
(SAV)

## Acknowledgments

We thank Peter Jebsen for valuable input on the evaluation of variables. The clinicians Olav Jetlund, Gunnhild Karevold, Åsa Karlsdottir, and Ellen Jaatun have contributed to the discussion and collecting of clinical data.

## Author Contributions

**Conceptualization:** Inger-Heidi Bjerkli, Lars Uhlin-Hansen, Elin Hadler-Olsen, Sonja E. Steigen.

**Data curation:** Inger-Heidi Bjerkli, Helene Laurvik, Elisabeth Sivy Nginamau, Tine M. Søland, Daniela Costea, Håkon Hov, Lars Uhlin-Hansen, Sonja E. Steigen.

**Formal analysis:** Sonja E. Steigen.

**Investigation:** Inger-Heidi Bjerkli.

**Methodology:** Helene Laurvik, Elisabeth Sivy Nginamau, Tine M. Søland, Daniela Costea, Håkon Hov, Lars Uhlin-Hansen, Elin Hadler-Olsen, Sonja E. Steigen.

**Supervision:** Lars Uhlin-Hansen, Elin Hadler-Olsen, Sonja E. Steigen.

**Writing – original draft:** Inger-Heidi Bjerkli.

**Writing – review & editing:** Inger-Heidi Bjerkli, Helene Laurvik, Elisabeth Sivy Nginamau, Tine M. Søland, Daniela Costea, Håkon Hov, Lars Uhlin-Hansen, Elin Hadler-Olsen, Sonja E. Steigen.

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
