## [Decision Letter · Decision Letter 0]

26 Jun 2020

PONE-D-20-15815

Tumor budding score predicts lymph node status in oral tongue squamous cell carcinoma and should be included in the pathology report

PLOS ONE

Dear Dr. Steigen,

Thank you for submitting your manuscript to PLOS ONE. After careful consideration, we feel that it has merit but does not fully meet PLOS ONE’s publication criteria as it currently stands. Therefore, we invite you to submit a revised version of the manuscript that addresses the points raised during the review process.

We look forward to receiving your revised manuscript.

Kind regards,

Marco Magalhaes, DDS, MSc, PhD, FRCDC

Academic Editor

PLOS ONE

Journal Requirements:

2. Please ensure you have thoroughly discussed any potential limitations of this study within the Discussion section.

3.Thank you for stating the following in the Acknowledgments Section of your manuscript:

'We thank Helse Nord (Nonprofit organization) for financial support. The publication charges for this article have been funded by a grant from the publication fund of Uit The arctic University of Norway.'

 'The authors received no specific funding for this work.'

Reviewers' comments:

Reviewer's Responses to Questions

**Comments to the Author**

1. Is the manuscript technically sound, and do the data support the conclusions?

Reviewer #1: Yes

Reviewer #2: Yes

2. Has the statistical analysis been performed appropriately and rigorously? 

Reviewer #1: No

Reviewer #2: Yes

3. Have the authors made all data underlying the findings in their manuscript fully available?

Reviewer #1: Yes

Reviewer #2: Yes

4. Is the manuscript presented in an intelligible fashion and written in standard English?

Reviewer #1: Yes

Reviewer #2: No

5. Review Comments to the Author

Reviewer #1: In this study, the authors aimed to evaluate the correlation among tumor budding (TB), lymph node metastases, tumor size (T 8 ed) and depth of invasion (DOI), and then assess their prognostic value in oral tongue squamous cell carcinoma. They concluded that T-classification (T 8 ed) and TB in a 3-tier score were independent prognostic factors. However, some concerns should be addressed before acceptance for publication in this journal.

1. Increasing evidences showed that tumor budding, DOI and combination of tumor budding with DOI are correlated with lymph node metastasis in oral tongue squamous cell carcinoma (OTSCC) and early stage OTSCC. It does not add new knowledge to the existing literature.

2. In page 8, line 146-150, the authors claimed that they combined tumor budding and DOI, but the following description was tumor thickness. Authors should state this clearly since the DOI is not same with tumor thickness.

3. As shown in Table 4, HRs values are beyond 95%CI. Authors should carefully check the statistical analysis.

4. Several studies showed that TB was a prognostic marker for early stage OTSCC, which was contradictory with the results in this study. The author should discuss it in the part of discussion.

Reviewer #2: This study involves taking a step back to assess what the T-status changes of the TNM8 mean for prognostication with respect to depth of invasion (DOI), as well as to assess tumour budding (TB), both together and separate from DOI in oral tongue SCC. The authors make a strong case for their title, which is the overall statement of part of the study findings. The authors don't include their assessment of the new system on DOI within the title, therefore they should consider altering the title to encompass this aspect of the study.

MAJOR comments:

Authors could consider writing a table which includes the current criteria for grading T and add their suggested inclusions in italics including the strict criteria of measuring budding.

Previous references that support including TB in T-grading using the high (>=5) or low (<5) buds/field. The authors here used the two and three tiered systems. They should comment based on their results which one seems more appropriate, and add it to the above mentioned Table.

Reference 17 refers to colorectal carcinoma TB? This seems an inappropriate reference to include as the entire paragraph is citing OTSCC

MINOR comments:

Abstract - a large number of grammatical errors. Please revise.

Page 2 background: “the majority of oral cavity arises in the oral tongue”. Should be “the majority of oral cavity cancers arise in the oral tongue”.

“where of depth of invasion” should be “where depth of invasion”

Methods:

Page 2 Line 31 “diagnosed with primary oral tongue squamous cell were” should be “diagnosed with primary oral tongue squamous cell carcinoma were”

Page 16 line 270: “The arctic University of Norway”. Should be capitalized

Page 5 line 81-82 “those who remains” should be “those who remain”

6. PLOS authors have the option to publish the peer review history of their article (what does this mean?). If published, this will include your full peer review and any attached files.

Reviewer #1: No

Reviewer #2: No

---

## [Author Response · Author response to Decision Letter 0]

13 Jul 2020

Response to reviewers 

Thank you for reviewing our manuscript “Tumor budding score predicts lymph node status in oral tongue squamous cell carcinoma and should be included in the pathology report” PONE-D-20-15815, and allowing us to submit a revised version. We appreciate the constructive comments made by the reviewers, which we have answered as specified below, point- by point. We hope you find the changes appropriate. 

 Responses to comments made by the academic editor:

1. PLOS ONE’s style requirements

We have ensured that the manuscript meets the PLOS ONE’s style requirements.

2. Please ensure that potentially limitations have been discussed

We have discussed the limitations of the study in the discussion. This includes the study being smaller than some previous studies. 

3. Acknowledgement and Funding statement

We have withdrawn the text about funding from the Acknowledgements section. Instead, we would like to change the funding statement to: “Inger-Heidi Bjerkli has received financial support for this work from Helse Nord (Nonprofit organization). The funder had no role in planning or execution of the study, nor in interpretation of results or writing of the manuscript.” 

Responses to comments made by reviewer 1:

1. Increasing evidences showed that tumor budding, DOI and combination of tumor budding with DOI are correlated with lymph node metastasis in oral tongue squamous cell carcinoma (OTSCC) and early stage OTSCC. It does not add new knowledge to the existing literature.

We agree that several studies have assessed the correlation between tumor budding, DOI and lymph node metastases previously. For prognostic markers to be included in clinical practice, their relevance must be validated in several independent patient cohorts. Thus, we believe the present study is important although other studies have reported similar results previously. Furthermore, the correlation with lymph node metastases is only one of several results reported in our manuscript. 

2. In page 8, line 146-150, the authors claimed that they combined tumor budding and DOI, but the following description was tumor thickness. Authors should state this clearly since the DOI is not same with tumor thickness.

We agree that the text is not precisely formulated. Almangush uses the term “tumor invasion” in the work with tumor budding and DOI (reference 14), and we have in our text used the term tumor thickness. We have indeed measured depth of invasion, and have corrected the revised manuscript accordingly.

3. As shown in Table 4, HRs values are beyond 95%CI. Authors should carefully check the statistical analysis.

We thank the reviewer for noting this mistake, which was due to inconsistent use of reference category. We have now corrected this in the manuscript. 

4. Several studies showed that TB was a prognostic marker for early stage OTSCC, which was contradictory with the results in this study. The author should discuss it in the part of discussion.

We agree that our results are not in line with several previous studies finding tumor budding to be a prognostic marker in low-stage OTSCC. The study by Amangush et al. (2015) and Xie et al (2015) included more patients with low-stage disease than our study did. Although we believe that our cohort is of reasonable size, our study may have been underpowered when we separate it into sub-groups. We have included this in the discussion and noted it to be a limitation of our study. Furthermore, the proportion of tumors with high-grade budding was lower in our study compared to the previous studies by Almangush and Xie. Few cases in the high-grade budding group could also inflict on the prognostic value. We have discussed this in the Discussion part.

Responses to comments made by Reviewer 2

MAJOR comments:

1. Authors could consider writing a table which includes the current criteria for grading T and add their suggested inclusions in italics including the strict criteria of measuring budding. 

We have considered writing a table including the current criteria for grading T as well as tumor budding. However, as we have not proposed to include tumor budding into a new TNM we find this difficult. We may have misunderstood the reviewer, and we are willing to look into this again if the reviewer can elaborate on this comment somewhat more. 

2. Previous references that support including TB in T-grading using the high (>=5) or low (<5) buds/field. The authors here used the two and three tiered systems. They should comment based on their results which one seems more appropriate, and add it to the above mentioned Table. 

None of the TB scoring systems were independent prognosticators of survival in multivariate analyses in our study. The correlation to lymph node metastases was stronger for the TB 3-tier system than for the 2-tier system. The HR in univariate survival analyses was larger for the TB 2-tier system than for the 3-tier system, but with a larger CI and higher p-value. Taken together these results suggest that the TB 3-tier system has better prognostic power than the 2-tier system, but our results do not allow definite conclusions to be drawn. We have included a paragraph discussing this in the Discussion part of the manuscript. 

3. Reference 17 refers to colorectal carcinoma TB? This seems an inappropriate reference in citing OTSCC

We have removed this reference in the revised manuscript.

MINOR comments:

1. Abstract - a large number of grammatical errors. Please revise.

We agree and apologize, and have revised the abstract as suggested.

2. Page 2 background: “the majority of oral cavity arises in the oral tongue”. Should be “the majority of oral cavity cancers arise in the oral tongue”.

“where of depth of invasion” should be “where depth of invasion”

3. Methods: Page 2 Line 31 “diagnosed with primary oral tongue squamous cell were” should be “diagnosed with primary oral tongue squamous cell carcinoma were” 

4. Page 16 line 270: “The arctic University of Norway”. Should be capitalized

5. Page 5 line 81-82 “those who remains” should be “those who remain”

We appreciate the linguistic corrections, and have corrected as suggested in the revised manuscript.

---

## [Decision Letter · Decision Letter 1]

1 Sep 2020

PONE-D-20-15815R1

Tumor budding score predicts lymph node status in oral tongue squamous cell carcinoma and should be included in the pathology report

PLOS ONE

Dear Dr. Steigen,

Thank you for submitting your manuscript to PLOS ONE. After careful consideration, the reviewers are supportive of publication and recommended a statistical review which was completed. The statistician raised important concerns that should be addressed before publication and are described in detail below. Therefore, we invite you to submit a revised version of the manuscript that addresses the points raised during the review process.

We look forward to receiving your revised manuscript.

Kind regards,

Marco Magalhaes, DDS, MSc, PhD, FRCDC

Academic Editor

PLOS ONE

Reviewers' comments:

Reviewer's Responses to Questions

**Comments to the Author**

1. If the authors have adequately addressed your comments raised in a previous round of review and you feel that this manuscript is now acceptable for publication, you may indicate that here to bypass the “Comments to the Author” section, enter your conflict of interest statement in the “Confidential to Editor” section, and submit your "Accept" recommendation.

Reviewer #1: All comments have been addressed

Reviewer #2: All comments have been addressed

Reviewer #3: (No Response)

2. Is the manuscript technically sound, and do the data support the conclusions?

Reviewer #1: Yes

Reviewer #2: (No Response)

Reviewer #3: Yes

3. Has the statistical analysis been performed appropriately and rigorously? 

Reviewer #1: I Don't Know

Reviewer #2: (No Response)

Reviewer #3: No

4. Have the authors made all data underlying the findings in their manuscript fully available?

Reviewer #1: Yes

Reviewer #2: (No Response)

Reviewer #3: No

5. Is the manuscript presented in an intelligible fashion and written in standard English?

Reviewer #1: Yes

Reviewer #2: (No Response)

Reviewer #3: No

6. Review Comments to the Author

Reviewer #1: The authors have addressed the concerns raised in the first round of review. However, we suggest that all statistical analyses should be checked by a statistician to make sure the accuracy.

Reviewer #2: (No Response)

Reviewer #3: The statistical analysis of the paper is torturing to read. I think the analysis is correct but the presentation makes it very hard to follow, even relatively simple topics. Please revise the whole analysis part and make the information you use more 'explicit'. The reader should not have to guess what has been done.

All results were considered significant -> The significance level for the hypothesis tests was 0.05.

All

167 results were considered significant if p≤0.05. -> Same here

Add citation for hypothesis testing:

https://www.mdpi.com/2504-4990/1/3/54

Table 1 is unclear. What is n^185? What are the individual rows?

Correlations analysis was 5-dimensional if I understand correctly? Be explicit.

variables TB,

191 DOI, T old, T 8 ed, and lymph node status (N).

Looking at table 2 I guess that is not true.

Write: The correlations has be calculated between what and what. Each vector had a length of what?

In the Methods section the authors talk about boostrap. In the Results section, I do not know where this has been used.

Univariate survival analysis: Please add Kaplan Meier Curves and discuss them in the text.

Multivariate survival analyses: This section is unclear to me. Are you performing a Cox Propoertional Hazard Model? Please revise the whole section.

7. PLOS authors have the option to publish the peer review history of their article (what does this mean?). If published, this will include your full peer review and any attached files.

Reviewer #1: No

Reviewer #2: No

Reviewer #3: No

---

## [Author Response · Author response to Decision Letter 1]

8 Sep 2020

Thank you for reviewing our manuscript “Tumor budding score predicts lymph node status in oral tongue squamous cell carcinoma and should be included in the pathology report” PONE-D-20-15815R1, and allowing us to submit a second revision of the manuscript. We appreciate the constructive comments made by the reviewers, which we have answered as specified below, point- by point. We hope you find the changes made appropriate. 

Responses to comments made by the academic editor:

We have no laboratory protocols as we have only done statisitcal analyses from our database.

Responses to comments made by reviewer 3:

1. All results were considered significant -> The significance level for the hypothesis tests was 0.05.

In the abstract and in the method part, under statistics (line 167): 

All results were considered significant if p ≤ 0.05 has been changed to: The significance level for the hypothesis test was 0.05.

2. Add citation for hypothesis testing: https://www.mdpi.com/2504-4990/1/3/54

The citation for hypothesis testing has been added to the materials and method section (statistical analyses).

3. Table 1 is unclear.

We agree that table 1 is unclear and have extended the heading. We hope that it now is more to the point:

Shift in T-classification; pTold in the first column, the pT8th in the second. Coincident numbers indicates no changes in T-classification while discrepancy indicates a shift from a lower to a higher classification, or from a higher to lower (only two cases). The third column indicates the number of cases for each of the alternative

4. Correlation analysis was not explained explicit enough, and it was not clear which vectors were in the correlation.

We agree that this was not explained explicit enough, and have rewritten the first sentence in the paragraph where the vectors are better presented:

Correlation analyses were performed on the whole cohort to evaluate if there was a statistical relationship between tumor budding (2 and 3 tier) and depth of invasion, T-classification (old and new) or lymph node status (N).

5. It is not clear for which calculations bootstrap has been performed.

We agree that this was not specified for each table. Under the material and method section (statistical analyses), we have now specified that we used bootstrapping when performing Spearman correlation analyses (table 2 and 3), and when applying Cox regression. This is in table 4 and 5.

6. Univariate survival analysis: Please add Kaplan Meier Curves and discuss them in the text.

We have added Kaplan Meier curves (figure 2) for budding, and have discussed this in the text (discussion part): 

The Kaplan Meier curves also demonstrates that the high budding group according to the 3-tire system has particularly poor survival, suggesting that patient in this group may require more extensive treatment and a closer follow-up than patients with low-budding tumors.

7. The multivariate section is unclear. 

We agree that this section could be clearer. We have rewritten parts of the section, and hope it is now more to the point. We have attached a manuscript file with all changes labeled (track changes) and one without mark-ups.

---

## [Editor Report · Decision Letter 2]

14 Sep 2020

Tumor budding score predicts lymph node status in oral tongue squamous cell carcinoma and should be included in the pathology report

PONE-D-20-15815R2

Dear Dr. Steigen,

We’re pleased to inform you that your manuscript has been judged scientifically suitable for publication and will be formally accepted for publication once it meets all outstanding technical requirements.

Kind regards,

Marco Magalhaes, DDS, MSc, PhD, FRCDC

Academic Editor

PLOS ONE
---

## [Editor Report · Acceptance letter]

16 Sep 2020

PONE-D-20-15815R2 

**Tumor budding score predicts lymph node status in oral tongue squamous cell carcinoma and should be included in the pathology report**

Dear Dr. Steigen:

I'm pleased to inform you that your manuscript has been deemed suitable for publication in PLOS ONE. Congratulations! Your manuscript is now with our production department. 

Kind regards, 

on behalf of

Dr. Marco Magalhaes 

Academic Editor

PLOS ONE